# Appropriateness of Antibiotic Prescribing for Acute Conjunctivitis: A Cross-Sectional Study at a Specialist Eye Hospital in Ghana, 2021

**DOI:** 10.3390/ijerph191811723

**Published:** 2022-09-17

**Authors:** Paa Kwesi Fynn Hope, Lutgarde Lynen, Baaba Mensah, Faustina Appiah, Edward Mberu Kamau, Jacklyne Ashubwe-Jalemba, Kwame Peprah Boaitey, Lady Asantewah Boamah Adomako, Sevak Alaverdyan, Benedicta L. Appiah-Thompson, Eva Kwarteng Amaning, Mathurin Youfegan Baanam

**Affiliations:** 1Bishop Ackon Memorial Christian Eye Centre, Cape Coast P.O. Box AD 184, Ghana; 2Institute of Tropical Medicine, 2000 Antwerp, Belgium; 3UNICEF/UNDP/World Bank/WHO, The Special Programme for Research and Training in Tropical Diseases (TDR), 1211 Geneva, Switzerland; 4Medwise Solutions, KNH, P.O. Box 2356, Nairobi 00202, Kenya; 5Institute for Evidence-Based Healthcare, Bond University, Gold Coast, QLD 4226, Australia; 6Council for Scientific and Industrial Research-Water Research Institute, Accra P.O. Box M 32, Ghana; 7Tuberculosis Research and Prevention Center, Yerevan 0014, Armenia; 8Cape Coast Teaching Hospital, Cape Coast CT 1363, Ghana; 9AngloGold Ashanti Ghana, Obuasi Mine, Obuasi P.O. Box 2665, Ghana; 10University Eye Hospital, University of Cape Coast, Cape Coast P.O. Box 5007, Ghana

**Keywords:** SORT IT, operational research, acute conjunctivitis, topical antibiotics, antimicrobial resistance, Ghana, AWaRe, West Africa, eye care

## Abstract

Most presentations of conjunctivitis are acute. Studies show that uncomplicated cases resolve within 14 days without medication. However, antibiotic prescription remains standard practice. With antimicrobial resistance becoming a public health concern, we undertook this study to assess antibiotic prescription patterns in managing acute conjunctivitis in an eye hospital in Ghana. We recorded 3708 conjunctivitis cases; 201 were entered as acute conjunctivitis in the electronic medical records (January to December 2021). Of these, 44% were males, 56% were females, 39% were under 5 years, 21% were children and adolescents (5–17 years) and 40% were adults (≥18 years). A total of 111 (55.2%) patients received antibiotics, of which 71.2% were appropriately prescribed. The use of antibiotics was more frequent in children under 17 years compared to adults (*p* < 0.0001). Of the prescribed antibiotics, 44% belonged to the AWaRe “Access” category (Gentamycin, Tetracycline ointment), while 56% received antibiotics in the “Watch” category (Ciprofloxacin, Tobramycin). Although most of the antibiotic prescribing were appropriate, the preponderance of use of the Watch category warrants stewardship to encompass topical antibiotics. The rational use of topical antibiotics in managing acute conjunctivitis will help prevent antimicrobial resistance, ensure effective health care delivery, and contain costs for patients and the health system.

## 1. Introduction

The conjunctiva is a transparent mucous membrane that covers the anterior part of the sclera up to the limbus as well as the inner part of the eyelids [1]. It is very rich in blood vessels and semi-transparent. When this tissue becomes inflamed, it is referred to as conjunctivitis. Conjunctivitis is a broad term that is used to refer to a diverse group of diseases that affects the conjunctival tissue [2]. This condition is often characterized by dilatation of the conjunctival blood vessels, which leads to reddening and swelling and is usually accompanied by some discharge [3]. It is the commonest presenting condition in eye care facilities globally [4]. It can be grouped into acute and chronic. Acute conjunctivitis is defined by the onset of a presenting complaint of 3 to 4 weeks while chronic conjunctivitis is more than 4 weeks in duration [5]. A study in Ghana identified acute conjunctivitis as the second most common cause of ocular morbidity [6]. There are two types of conjunctivitis, infectious and non-infectious [7]. Viral conjunctivitis is the most common cause of infectious conjunctivitis followed by bacterial conjunctivitis [4]. Non-infectious conjunctivitis could be caused by allergies, trauma or chemicals. Bacterial conjunctivitis is characterized by moderate to severe redness, mucopurulent discharge and occasional matting of eyelids. Patients with viral conjunctivitis also present with mild to moderate redness and watery discharge but little to no matting of the eyelids. Allergic conjunctivitis typically presents with mild to moderate redness, itching and tearing [4].

Often, acute conjunctivitis cases are managed with topical antibiotics, although studies suggest that this is not necessary in most cases [8]. Systematic reviews have shown that most acute conjunctivitis cases are self-limiting and can resolve without topical antibiotics [9]. With the growing global public health concern regarding antimicrobial resistance (AMR), antibiotic stewardship is relevant in the management of conjunctivitis [10].

Data from low-and middle-income countries (LMICs) confirm that antimicrobial-resistant bacterial infections are common [11]. A systematic review of studies from West Africa shows a high risk of AMR compromising first-line empirical treatment [12]. In Ghana, studies found that a high percentage of bacterial isolates are resistant to commonly used antibiotics [13,14]. According to the World Health Organization (WHO), there is an estimated 80% of antibiotics being used in the community and of this figure, 20–50% are inappropriately used [15]. A study in Mozambique identified excessive and inappropriate use of antibiotics as a major setback in the fight against antimicrobial resistance [16]. Another study in Nigeria recommended antimicrobial stewardship interventions to address the excessive use of antibiotics and promote appropriate prescribing [17].

The fourth pillar of the National AMR policy (adopted from the WHO document) advocates for the optimal use of antimicrobial agents [18]. Topical antibiotics are not listed in the AWaRe classification and are not considered in the WHO practical toolkit for antibiotic stewardship programs [19,20]. However, their overuse contributes to the development of antibiotic resistance [21]. Moreover, the practice of prescribing unnecessary topical antibiotics adds to the cost of management which will be borne either directly by the patient or indirectly through the National Health Insurance Scheme. In LMICs, bacterial illnesses come with a higher economic burden as a result of the limited availability of second-line antibiotics [22]. In undertaking this study, we sought to draw attention to the gap identified in the non-addressing of the topical antibiotic stewardship program. We are convinced this is indeed a prime area that needs prompt attention to prevent antimicrobial resistance in eye care. Added to this is the possibility of patients who could in later life require surgeries for ageing conditions such as cataracts. In the event that antimicrobial resistance is developed through unwarranted prescriptions of antibiotics in the management of acute conjunctivitis, the healing process of post-cataract surgeries could be impeded, leading to far more damaging consequences such as endophthalmitis and ultimately blindness.

More rational use of topical antibiotics will help the health system reduce costs by eliminating unnecessary expenditure as part of the management protocols for acute conjunctivitis. This study aimed at documenting prescribing practices and assessed the appropriateness of current antibiotic prescriptions in a specialist eye hospital in Ghana. It also served as baseline data for determining patient and prescriber characteristics associated with antibiotic prescription patterns. Our findings would help in policy integration of topical antibiotics.

## 2. Materials and Methods

### 2.1. Study Design

A retrospective cross-sectional study was carried out among patients presenting with acute conjunctivitis, using routine data extracted from electronic medical records and complemented with data from the patient folders.

### 2.2. General Setting

Ghana is located in West Africa with a population of 30.8 million as of 2021. The capital city is Accra. Health care services are offered by both the private and public sectors. The private sector health facilities are owned and managed by private individuals and institutions. The public sector facilities are managed directly by the Ministry of Health and indirectly by agencies under the ministry, namely the Ghana Health Service (GHS) and the Christian Health Association of Ghana (CHAG). Health services are accessed either through health insurance schemes (both private and public) or out-of-pocket cash payments.

In the Ghanaian context, optometrists and ophthalmic nurses are mostly the first point of call in the primary eye care facilities as well as pharmacists at the community level (in cases of over-the-counter purchases). The ophthalmologists are mainly based at secondary and tertiary health care facilities. Although certain public health care facilities are graded as secondary and tertiary facilities, they still receive and attend to patients with primary eye care needs. Most community-level facilities do not have eye care professionals (ophthalmologists, optometrists or ophthalmic nurses). As a result, they are guided strictly by the 7th Edition of the Ghana Standard Treatment Guidelines (STG) for their management of conjunctivitis. The current practice is to manage conjunctivitis with antibiotic eye drops and/or ointment in the case of bacterial origin, with mast cell stabilizers in the case of allergic conjunctivitis, and symptomatic management in the case of viral conjunctivitis. The approved medication list at the community level for the treatment of conjunctivitis is Tetracycline 1% ointment, Chloramphenicol eye drops 0.5% or Ciprofloxacin eye drops 0.3% [23].

However, in facilities with eye care professionals, permission is granted to prescribe additional ophthalmic medications as deemed appropriate by the level of clearance given by the National Health Insurance Authority (NHIA) to that facility. These medications are aligned to the scope of practice of the various eye cadres.

### 2.3. Specific Setting

Bishop Ackon Memorial Christian Eye Centre (BAMCEC) is a specialist (eye) hospital that offers eye care services to a wide range of patients in and outside the Cape Coast Metropolis. BAMCEC records an average of 17,000 outpatient attendants per year. The services range from primary eye care (walk-in patients and first-timers) to tertiary eye care services (which include cases that have been referred from other facilities). The facility is also a training center for optometry and ophthalmic nursing students in their clinical years as well as student-opticians. They have an average outpatient department (OPD) attendance of 100 patients a day and a workforce of 55 staff members. This includes a core eye care staff of two part-time ophthalmologists, four optometrists, five ophthalmic nurses and eight opticians who are all full-time staff. They also have one pharmacist. All three cadres (ophthalmologists, optometrists and ophthalmic nurses) have consulting as part of their job description. The optometrists and ophthalmic nurses, however, see most of the first-timers and only refer the complex cases to the ophthalmologists. In addition to the approved list of topical antibiotics at the community level, BAMCEC has clearance for all levels of ophthalmic antibiotic medications (aminoglycosides, fluoroquinolones and polymyxin B combinations) and all cadres are permitted to prescribe them.

The hospital employs an electronic medical system in addition to the traditional hard copy patient folders. All details of the patient from the records office to the last prescriber (ophthalmologist, optometrist or ophthalmic nurse) are fully written in the folders. The prescriber then logs onto the electronic medical system. This is an application software for the hospital which has all the diagnoses, the medications in use and their various dosages pre-entered into the system. The prescriber only needs to drop down the list, choose the diagnosis and add the management plan as desired.

### 2.4. Participants

All patients with new acute conjunctivitis who presented to BAMCEC from 1 January to 31 December 2021, were included in the study. The list of all cases that had been diagnosed and entered as acute conjunctivitis was generated from the electronic medical record system. The corresponding hard copy folders were retrieved from the records office for data collection. As per the working definition at the hospital, cases of conjunctivitis that had an onset of presentation of 14 days or less were considered acute. All others that had exceeded 14 days were regarded as chronic.

### 2.5. Variables

These included: demographics (age, sex), antibiotic prescription (yes/no) and type, antibiotic combinations (yes/no) and type of combination and eye care staff cadre that attended to each case. For all patients for whom antibiotics were prescribed, the appropriateness of antibiotic prescription was assessed.

### 2.6. Data Collection

Data were collected by five staff of BAMCEC; one ophthalmologist, two optometrists, one ophthalmic nurse and one general nurse. All data were collected between March and April 2022. The appropriateness of antibiotic use was decided by two groups from the same data collection team. The folders that contained prescribed antibiotics were reviewed by a team of two (one ophthalmologist and one optometrist) who cross-checked the presenting complaints and symptoms per chart and grouped them into two (bacterial and non-bacterial). When a patient showed signs of mucopurulent discharge with moderate to severe erythema, this would indicate acute conjunctivitis of bacterial origin. A second team (one optometrist and one ophthalmic nurse) reviewed the two categories (bacterial and non-bacterial) that had been identified by the first. For all the cases of bacterial origin, prescribing antibiotics was deemed appropriate. For the cases considered of non-bacterial origin, prescribing antibiotics was deemed as not appropriate.

### 2.7. Statistical Analysis

Data were entered in Excel and analyzed in EpiData analysis software (version 2.2.3.187, EpiData Association, Odense, Denmark). Demographic characteristics, patterns of antibiotic prescriptions and the appropriateness of the use of antimicrobial medications were summarized as proportions. The chi-square test was used to assess for differences in proportions between age groups and staff cadres. Statistical significance was set at 5% (*p* < 0.05).

## 3. Results

From 1 January to 31 December 2021, 3708 cases of conjunctivitis presented at the hospital. Of these, 201 (5%) were recorded as acute conjunctivitis in the electronic medical records. Table 1 shows the socio-demographic characteristics of patients presenting with acute conjunctivitis. The majority of patients were under-fives or adults (>=18-years-old) and females were preponderant.

Table 2 and Figure 1 show the results with reference to the antibiotic prescription patterns. More than half of the 201 cases received topical antibiotic prescriptions. The most frequently prescribed antibiotic was Gentamycin eye drops, and 54% of patients received a topical antibiotic belonging to the “Watch” classification. Approximately one-fifth (21) of the patients received a combination of an eye ointment and drops. The most frequently prescribed eye ointment in combination was Tetracycline (57.1%).

Over two-thirds of the patients (71.2% (95%CI 61.8, 79.4) prescribed antibiotic medications were adjudged to have been appropriately prescribed.

Most of the prescriptions were given by optometrists and ophthalmic nurses.

Proportions for each of the AwaRe class calculated on total number of antibiotic prescribed as the denominator. 

Figure 2 shows the number and proportion of patients who received a prescription of topical antibiotics.

Table 3 shows the factors that were examined for a possible association with antibiotic prescription. Children and adolescents were more likely to receive antibiotics than adults (*p* < 0.0001). No significant difference was observed between male and female patients or between the different staff cadres.

Figure 3 shows antibiotic prescribing practice for conjunctivitis per age group. Children received more antibiotics than adults.

Table 4 shows the factors that were examined for a possible association with appropriate antibiotic prescription. No significant association was found between age, sex, attending cadre and the appropriateness of the antibiotic prescription.

## 4. Discussion

This study sought to describe the antibiotic prescription patterns in the treatment/management of acute conjunctivitis in a specialist health facility that also provides primary care services to the catchment area. Among the 3708 cases of conjunctivitis presenting to the eye clinic between January and December 2021, 201 (5%) were labelled as acute conjunctivitis. Over 50% of patients received a topical antibiotic, mainly Gentamycin eye drops, and for 71% of the prescriptions, this was considered appropriate. However, 54% of the prescriptions contained an antibiotic from the “Watch” class.

Although the 71.2% score on the appropriateness of the antibiotic prescription was quite favorable [15], a relatively high proportion of antibiotics used were from the Watch category. The Access, Watch, Reserve (AWaRe) classification of antibiotics was developed to guide antibiotic stewardship at local, national and global levels, with relatively more emphasis on sparing the use of the WATCH and RESERVE antibiotics [24]. In our study in BAMCEC, two frequently used topical drugs (Tobramycin and Ciprofloxacin) are considered as WATCH category, while Tetracycline and Gentamycin are in the ACCESS category. The WHO 13th General Programme of Work 2019–2023 recommends a country target of 60% of antibiotics prescribed from the ACCESS group. In our study, only 46% of the topical antibiotics prescribed belonged to this group. Although topical antibiotics are not listed in the AWaRe list or as a target in the Point Prevalence Survey (PPS) on antibiotic use [25], their widespread use and potential overuse will contribute to the development of antibiotic resistance [21]. This calls for monitoring topical antibiotic use as part of a comprehensive antibiotic stewardship program.

The majority of cases of acute conjunctivitis occurred in females and children. Two other studies from Ghana, conducted in eye clinics, found the same female preponderance, but the rate of acute conjunctivitis among all eye conditions ranged from 30% to 40% [6,26]. The preponderance of females and younger age groups may be indicative of the higher attendance of females and children at health care facilities. Our study showed a much lower rate of acute conjunctivitis among all cases presenting at the eye clinic. This can partially be explained by the fact that the case definition of acute conjunctivitis at BAMCEC was different. The two studies in Ghana did not specify the definition for acute conjunctivitis [6,26] but others have proposed a cut-off of 4 weeks to distinguish between acute and chronic conjunctivitis [27,28,29,30]. Another possibility for the lower rate of acute conjunctivitis recorded in our study is that as a referral center, more chronic cases were seen. However, the data from two other eye clinics in Ghana showed very different results. The published literature generally indicates that most presenting cases of conjunctivitis tend to be acute rather than chronic [4]. Therefore, possible misclassification while selecting the diagnosis in the electronic medical records is a more likely explanation. To allow for benchmarking and overall monitoring, correct case definitions or diagnostic categories must be used.

In our study, we observed that 55.2% of patients symptomatically diagnosed with acute conjunctivitis received antibiotics (as shown in Figure 2), which is lower than the 80% observed in another study in the Netherlands among primary care physicians [31]. Based on the fact that only 30–50% of clinically suspected bacterial conjunctivitis cases were confirmed by culture in this study, 80% was considered suboptimal management [31,32]. Our 55% use of antibiotics in acute conjunctivitis is more in line with an expected 50% of possible bacterial etiology, warranting treatment [31,32]. This observation indicates adherence to the established routine management protocol for acute conjunctivitis at the hospital.

It was also observed that in more than 70% of the patients, antibiotic prescription was considered appropriate. Although there is room for improvement, this suggests effective antimicrobial stewardship on the part of the prescribers at the hospital. A similar pattern was found in a study from South Africa where antibiotics were prescribed only for patients who needed them [33] as opposed to studies that highlighted over-prescription in the United Kingdom [34] and the Netherlands [31,32].

This study found that a minority of patients (one-fifth) classified as suffering from acute conjunctivitis received a prescription of antibiotic combinations. Having a minimal proportion of patients on combination therapy reinforces the aim to limit the prescription of antibiotics. The remaining 80% were on a single drug prescription which is in line with recommended protocols to reduce the risk of AMR in the future [35].

As expected, the ophthalmologists attended to only 1% of the cases of acute conjunctivitis. Ophthalmologists attend to more complex eye conditions in the hospital. No difference was observed between the staff cadres in terms of antibiotics prescribed and their appropriateness. This could be because of frequent clinical meetings amongst all prescribing cadres in the hospital. These meetings serve as fora for reviewing and agreeing on management protocols for conditions presented at the hospital. All cadres are thus applying similar standards to their case management. This contrasts with a study in New Zealand that looked at the influence of training level in general practitioners’ registrars on prescribing patterns. They found that established general practitioners prescribe antibiotics for conjunctivitis above guideline recommendations (74%), but prescribing rates are lower in later training [20]. A second study from the US showed that 68% of patients who visited a physician at an emergency room received antibiotic eye drops while this figure dropped to 36% for those who saw an ophthalmologist [36]. In our study, a high proportion of children under 5 years received antibiotics (92.3%) and 78% was considered appropriate. This is in line with the literature that states that while viral conjunctivitis is more frequent in adults, children have more bacterial conjunctivitis [37].

The study has the following strength. It is the first study in Ghana to describe the appropriate use of antibiotics in patients presenting with acute conjunctivitis. This will serve as a reference point for future surveys and as a basis for further research work in this area.

The study has the following limitations. The facility used for this study is a specialist facility; as such, patients who came in usually did so a bit late, this is after they have already attended some primary eye care facilities, and this contributed to a higher count of chronic cases and a relatively lower number of cases of acute conjunctivitis.

The small number of cases of acute conjunctivitis entered into the electronic system makes it difficult to extrapolate results to other prescribing healthcare facilities. The appropriateness of antibiotic prescriptions was based on the judgement of members of the same eye clinic team. No microbiological data were available to confirm or not the presence of bacterial conjunctivitis. However, this reflects the reality in most eye clinics, and files were assessed by two independent teams. The clinical diagnosis of bacterial conjunctivitis is not very reliable, but the 50% rate of antibiotic prescribing in our study suggests that there is at least not a big problem of over-prescription. Data were also only available for patients who had received antibiotics. This is a drawback; as inappropriate use of antibiotics also includes the non-prescription of antibiotics in patients with bacterial conjunctivitis. Future studies should include all patients, regardless of whether antibiotics were prescribed.

## 5. Conclusions

Antibiotic prescribing practice in BAMCEC is to a large extent appropriate and follows standard treatment guidelines. However, a relatively large proportion of antibiotics are from the Watch category, and still, almost 30% of the prescriptions were considered inappropriate. There is a need to further investigate the reasons underlying the inappropriate prescription of antibiotics in acute conjunctivitis. This study will serve to improve the management protocols for acute conjunctivitis to avoid antimicrobial resistance and to ensure sustainable eye care delivery. Because of their widespread use in the eye clinic, antibiotic stewardship programs should include the use of topical antibiotics, especially those from the “Watch” category. This also requires a standardized and correct case definition of acute conjunctivitis across all health service providers.

## Figures and Tables

**Figure 1 ijerph-19-11723-f001:**
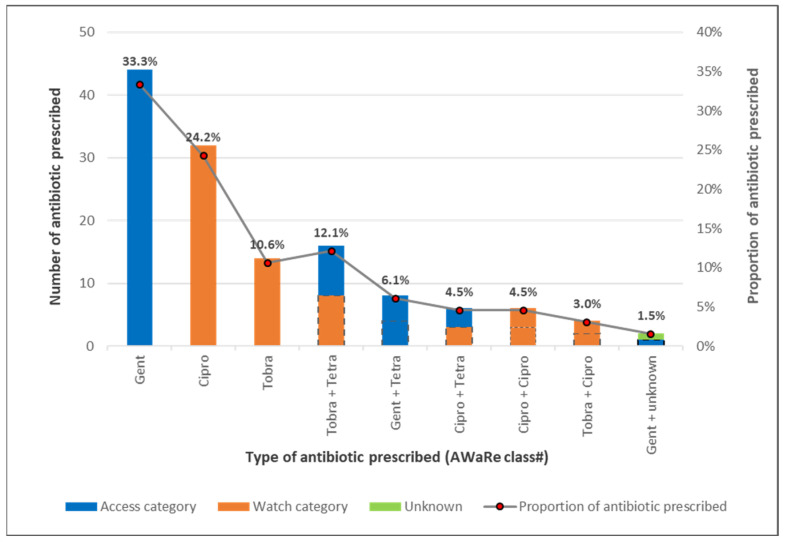
Antibiotic prescription pattern per WHO Access, Watch, Reserve (AWaRe) classification among patients presenting with acute conjunctivitis at a specialist eye hospital in Ghana from 1 January to 31 December 2021.

**Figure 2 ijerph-19-11723-f002:**
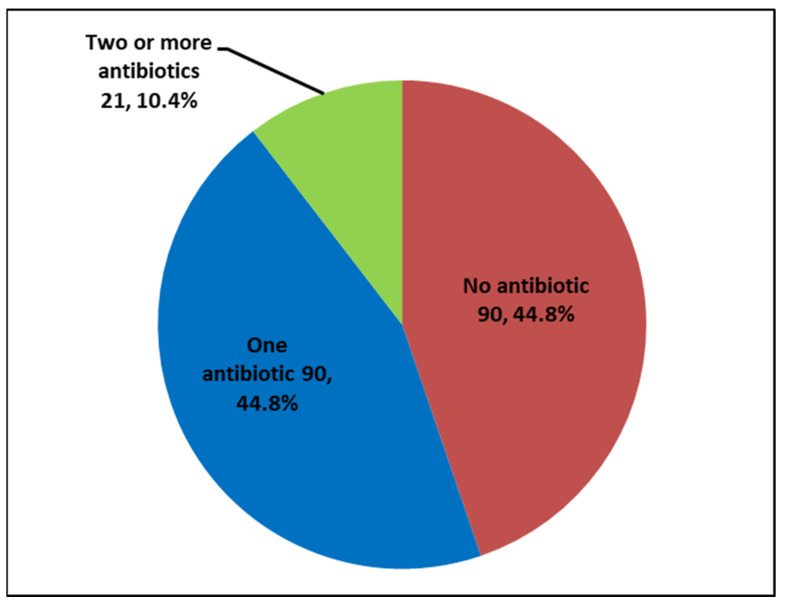
Number and proportion of patients who received a prescription of topical antibiotics.

**Figure 3 ijerph-19-11723-f003:**
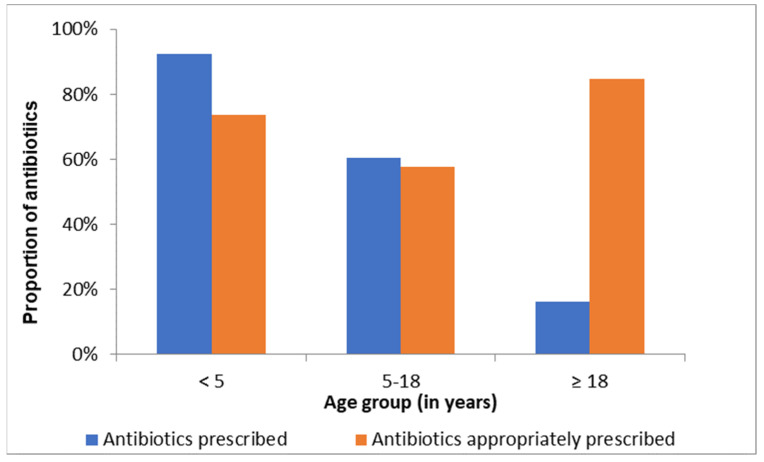
Proportions of antibiotics prescribed per age group.

**Table 1 ijerph-19-11723-t001:** Socio-demographic characteristics of patients presenting with acute conjunctivitis at a specialist eye hospital in Ghana from 1 January to 31 December 2021.

Variable	Patients with Acute Conjunctivitis n (%)
Age (Years) (n = 201)	
<5 years	78 (39)
5–17 years	43 (21)
≥18 years	80 (40)
Sex (n = 201)	
Male	89 (44)
Female	

**Table 2 ijerph-19-11723-t002:** Antibiotic prescription practices in a specialist eye hospital in Ghana from 1 January to 31 December 2021.

Variable	Number of Patients n (%)
**Antibiotic prescribed (n = 201)**	
Yes	111 (55.2)
No	90 (44.8)
**Attending Cadre (n = 201)**	
Ophthalmologist	2 (1.0)
Optometrist	107 (53.2)
Ophthalmic nurse	92 (45.8)
**Appropriate antibiotic prescription (n = 111)**	
Yes	79 (71.2)
No	32 (28.8)

**Table 3 ijerph-19-11723-t003:** Factors associated with antibiotic prescriptions in a specialist eye hospital in Ghana from 1 January to 31 December 2021.

Variable	Acute Conjunctivitis Cases	Antibiotics Prescription			*p*-Value
	N	Yes (n = 111)	No (n = 90)	Total (n = 201)	
**Age groups (in years)**	201				<0.0001 ^c^
<5 years		72 (92.3%)	6 (7.7%)	78 (100%)	
5–17 years		26 (60.5%)	17 (39.5%)	43 (100%)	
≥18 years		13 (16.2%)	67 (83.8%)	80 (100%)	
**Sex**	201				0.214 ^c^
Male		54 (60.7%)	35 (39.3%)	89 (100%)	
Female		57 (50.9%)	55 (49.1%)	112 (100%)	
**Cadre**	201				0.843
Ophthalmologist|Optometrist		59 (54.1%)	50 (45.9%)	109 (100%)	
Ophthalmic nurse		52 (56.5%)	40 (43.5%)	92 (100%)	

^c^ Pearson’s chi-square test with Yates’ continuity correction.

**Table 4 ijerph-19-11723-t004:** Factors associated with appropriate antibiotic prescriptions in a specialist eye hospital in Ghana from 1 January to 31 December 2021.

Variable	Prescribed Antibiotics Cases	Appropriate Prescription			*p*-Value
	N	Yes (n = 79)	No (n = 32)	Total (n = 111)	
**Age groups (in years)**	111				0.164 ^f^
<5 years		53 (73.6%)	19 (26.4%)	72 (100%)	
5–18 years		15 (57.7%)	11 (42.3%)	26 (100%)	
≥18 years		11 (84.6%)	2 (15.4%)	13 (100%)	
**Sex**	111				0.696 ^c^
Male		37 (68.5%)	17 (31.5%)	54 (100%)	
Female		42 (73.7%)	15 (26.3%)	57 (100%)	
**Attending Cadre**	111				0.526 ^f^
Ophthalmologist/Optometrist		44 (74.6%)	15 (25.4%)	59 (100%)	
Ophthalmic nurse		35 (67.3%)	17 (32.7%)	52 (100%)	

^c^ Pearson’s chi-square test with Yates’ continuity correction. ^f^ Fisher’s exact test.

## Data Availability

The dataset used is available upon request from the corresponding author.

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
