# Peer review of "Appropriateness of Antibiotic Prescribing for Acute Conjunctivitis: A Cross-Sectional Study at a Specialist Eye Hospital in Ghana, 2021"

_ijerph, 2022, doi:10.3390/ijerph191811723_

Round 1
Reviewer 1 Report
Please, check the attache file.

Reviewer 2 Report
Here, the authors describe a cross-sectional study of acute conjunctivitis presentations at a hospital in Ghana in 2021, and the treatment thereof. Of 201 patients, 55.2% were treated with antibiotics, of which 71.2% were classified as correctly prescribed (bacterial vs viral or other). Topical antibiotics were more commonly used in younger patients. In comparing those antibiotics classified as routine vs restricted, there was a slight predisposition to treating with the more restricted Watch category medications.
Thanks for inviting me to review this interesting article. I have a few comments which I hope the authors might consider. I would be amenable to reviewing a revised iteration.
I might suggest logistic regression would be a more appropriate analysis method than just Chi-square test. This would enable you to estimate measures of association (prevalence ratios) for appropriate vs inappropriate treatment, rather than just a p-value. Depending if you have data on whether it conjunctivitis aetiology was classified as likely bacterial, viral, or allergic, or other, you could apply multinomial logistic regression to evaluate treatment likelihood.
I note in the Results that 3708 cases of conjunctivitis were recorded, of which only 201 were classified as acute conjunctivitis. Does this imply that 3507 were chronic or just that it wasn’t specified whether it was acute or chronic? I think it is more likely that most of the 3708 were actually acute conjunctivitis, wouldn’t you think? If so, you could evaluate analyses of all 3708 and the treatment thereof, with a sensitivity analysis limited to the 201 which were definitively labeled as acute conjunctivitis. If the authors might consider this, that would enhance the article. If, however, most of the 3507 records not clearly labeled as acute conjunctivitis were also missing key demographic and clinical information, including the mode of treatment, then their exclusion would be justified. However, their exclusion should be clearly explained as due to their having too much missingness in the record, not just that only the 201 were considered likely acute conjunctivitis.
Please use sex and male/female terminology, not gender and men/women.
Round 2
Reviewer 1 Report
Check the attached file carefully.

Round 3
Reviewer 1 Report
Accept